# An Experimental Study on Trajectory Tracking Control of Torpedo-like AUVs Using Coupled Error Dynamics

Gun Rae Cho [1,*], Hyungjoo Kang [1], Min-Gyu Kim [1], Mun-Jik Lee [1], Ji-Hong Li [1], Hosung Kim [2], Hansol Lee [2] and Gwonsoo Lee [3]

1 Korea Institute of Robotics and Technology Convergence, Pohang 37666, Republic of Korea ; hjkang@kiro.re.kr (H.K.); zxdwa0817@kiro.re.kr (M.-G.K.); mcklee@kiro.re.kr (M.-J.L.); jhli5@kiro.re.kr (J.-H.L.)
2 Hanwha Systems, Gumi 39376, Republic of Korea; hosung0608.kim@hanwha.com (H.K.); hansol.lee@hanwha.com (H.L.)
3 Department of Mechatronics Engineering, Chungnam National University, Daejeon 34134, Republic of Korea; kali55@o.cnu.ac.kr
* Correspondence: sandman@kiro.re.kr; Tel.: +82-54-279-0459

**Abstract:** In this paper, we propose a trajectory tracking controller with experimental verification for torpedo-like autonomous underwater vehicles (AUVs) with underactuation characteristics. The proposed controller overcomes the underactuation problem by designing the desired error dynamics in a coupled form using state variables in body-fixed and world coordinates. Unlike the back-stepping control requiring high-order derivatives of state variables, the proposed controller only requires the first derivatives of the states, which can alleviate noise magnification issues due to differentiation. We adopt time delay estimation to estimate the dynamics indirectly using control inputs and vehicle outputs, making the proposed controller relatively easy to apply without requiring the all of the vehicle dynamics. We also address some practical issues that commonly arise in experimental environments: handling measurement noises and actuation limits. To mitigate the effects of noise on the controller, a filtering technique using a moving window average is employed. Additionally, to account for the actuation limits, we design an anti-windup structure that takes into consideration the nonlinearity between the thrusting force and rotating speed of the thruster. We verify the tracking performance of the proposed controller through experimentation using an AUV. The experimental results show that the 3D motion control of the proposed controller exhibits an RMS error of 0.3216 m and demonstrate that the proposed controller achieves accurate tracking performance, making it suitable for survey missions that require tracking errors of less than one meter.

**Keywords:** autonomous underwater vehicles; robust trajectory tracking; coupled desired error dynamics; time delay estimation



## 1. Introduction

Designing a trajectory tracking controller for torpedo-like autonomous underwater vehicles (AUVs) is challenging due to their underactuation characteristics. These vehicles have only three control inputs—surge force, pitch, and yaw moment—to control their 3D motion in space. The lack of control inputs leads to dissatisfaction of the matching condition, where certain uncertain terms in the state equation cannot be directly compensated for by the control inputs [1,2]. Additionally, the vehicles have nonlinear dynamics involving both rigid body dynamics and hydrodynamics [2]. In the development of AUVs and their control systems, it is crucial to verify their performance in experimental environments. During such verification, numerous issues can affect the system's performance, including sensor measurement noise, modeling errors in system dynamics, disturbances, and imperfections in control systems, such as jitter in the sampling time. As a result, developing

a trajectory tracking controller for AUVs is challenging in two aspects: the design of the control algorithm and the experimental verification process.

Regarding the controller design manner, there have been several research works to propose trajectory tracking control scheme for AUVs. To overcome the matching condition issues of the vehicles, several research works have proposed control schemes based on back-stepping control (BC). BC using the vehicle dynamic model was proposed in [3–6]. BC with time delay estimation (BCTDE), an indirect estimator of the vehicle dynamics, has also been studied [2,7,8]. BC provides an excellent and systematic method to handle matching condition issues. However, it requires high-order derivatives of the state variables, which may result in instability in experimental environments due to the magnification of noise effects. There have been several studies on controlling the vehicles using other schemes. Hierarchical design of the controllers has been researched to address the underactuation characteristics [9,10]. Sliding mode control has been applied to gain robustness against model errors and disturbances [11–14]. Adaptive schemes have been employed to resolve model uncertainty [15,16]. Neural networks have been used for robust path following [17,18].

Regarding the experimental verification manner, however, it is hard to find previous research works proposing trajectory tracking algorithms with experimental results. For example, the aforementioned previous research works primarily demonstrate control performance with simulation results and lack experimental verification. The reasons for the absence of experiments in previous works are not explicitly mentioned, but this could be attributed to the need for further investigation to address practical issues such as sensor noises or modeling errors in AUV dynamics, including disturbances. For example, the authors have attempted to verify the performance of the BCTDE [2,7,8] and found it difficult to determine stable gains for the BCTDE. Figure 1 illustrates the experimental results for depth control using the BCTDE, which indicate unstable responses. This instability arises because the BCTDE requires high-order differentiation of the state variables to handle unmatched dynamics and disturbances, thereby amplifying the effects of noise in the state measurements. As another reason for the lack of experimentation, the difficulty of obtaining a suitable experimental platform can be considered, as AUVs are costly plaforms. In constrast to the trajectory tracking problem, however, there have been several research works that proposed via point tracking control of underactuated AUVs with experimental verification [19–21]. These research works utilize traditional approaches employing PID-type controllers for the forward velocity, pitch angle (or depth), and heading angle, combined with a desired heading angle planner such as the line of sight (LOS) [22–24]. Designing via point tracking controllers is relatively straightforward since each controller focuses only on stabilizing states with dynamics matched to the control inputs. However, they mainly focus on waypoint tracking and are unable to handle time-varying trajectory tracking problems. There have also been research works suggesting trajectory tracking controllers for underwater vehicles with full degree-of-freedom (DOF) actuation [25–29]. In such cases, the vehicle can generate the control actuations for every controlled state, eliminating issues arising from a lack of satisfaction of the matching condition.

In this paper, we propose a robust trajectory tracking controller for the 3D motion of underwater vehicles, along with experimental verification of its control performance. In terms of controller design, the proposed controller incorporates an appropriate design of the desired error dynamics and time delay estimation (TDE). To address the underactuation issues of the vehicle, the desired error dynamics is formulated in a coupled form between the state variables in body-fixed and world coordinates. By utilizing the TDE [2,30], the controller effectively compensates for the nonlinear dynamics and disturbances of the vehicle while maintaining a simple structure. The proposed controller only requires the state variables and their first derivatives, mitigating the issues of noise amplification due to differentiation. An initial version of the proposed controller was presented in [31], and in this paper, we extend the controller for motion control in a 3D space. In terms of experimental implementation, practical issues related to measurement noise and actuation limitations

are addressed. A moving average filter is employed to mitigate the effects of sensor noise on control performance, and actuation limitations combined with nonlinear dynamics are also considered. We verify the tracking performance of the proposed controller through experiments conducted on an AUV.

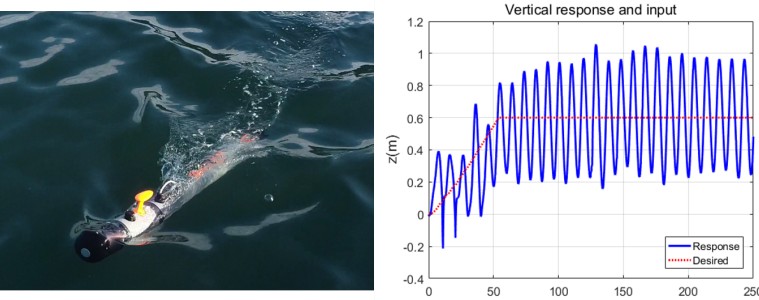

**Figure 1.** Experimental results for depth control using the BCTDE. The results indicate an almost unstable response due to the high−order differentiation of states required by the controller, which amplifies the noise effect in the state measurements.

## 2. Controller Design for the AUV

### 2.1. AUV Systems and Motion-Governing Equations

Figure 2 illustrates the AUV platform utilized in this research, which was developed by Hanwha Systems [32]. The AUV serves as a testbed for underwater docking tasks [33]. The linear velocities of the vehicle are measured using a Doppler velocity log (DVL), and the angular velocities are measured using an inertial measurement unit (IMU). The position of the vehicle in the world coordinate is estimated using an extended Kalman filter (EKF) that utilizes navigation sensor data from an IMU, a DVL, a depth sensor, a digital compass, and a global navigation satellite system (GNSS) [34–36]. The vehicle is equipped with a thruster for forward propulsion, as well as rudder fins and stern fins for lateral and vertical moments, respectively. The actuators are controlled by an ARM-based embedded system with a control sampling frequency of 10 Hz.

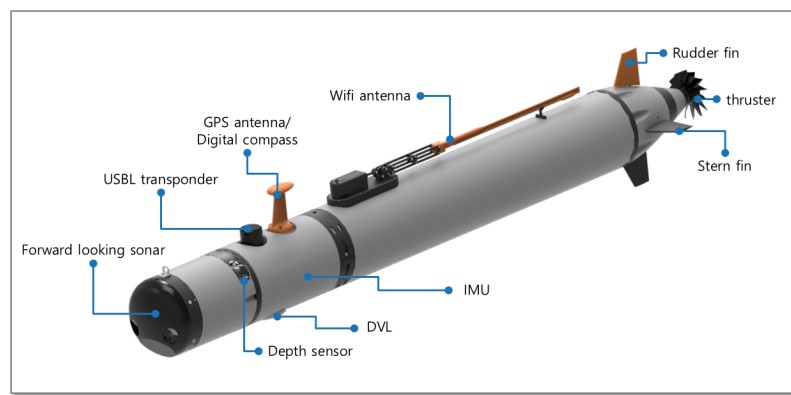

**Figure 2.** AUV used in the experiment. The vehicle was developed by Hanwha Systems [32]. The AUV serves as a testbed for underwater docking tasks [33].

To formulate the motion-governing equation, let us consider the control problem of the vehicle in a 3D space as shown in Figure 3. Assuming that the roll motion of the vehicle can be neglected, the governing equations for motion are given as follows [8,37]:

$$\dot{\boldsymbol{\eta}} = \mathbf{R}\boldsymbol{\nu}$$
$$\dot{\theta} = q,$$
$$\dot{\psi} = r/c\theta, \quad and,$$
(1)

$$
\begin{aligned}
m_{11}\dot{u} - m_{22}vr + m_{33}wq + f_u(u)u + \tau_{eu} &= \tau_u, \\
m_{22}\dot{v} + m_{11}ur + f_v(v)v + m_{22}\tau_{ev} &= 0, \\
m_{33}\dot{w} - m_{11}uq + f_w(w)w - d_1 + \tau_{ew} &= 0, \\
m_{55}\dot{q} - (m_{33} - m_{11})uw + f_q(q)q + (d_2 + \tau_{eq}) &= \tau_q, \\
m_{66}\dot{r} - (m_{11} - m_{22})uv + f_r(r)r + \tau_{er} &= \tau_r,
\end{aligned}
\tag{2}
$$

where $c\bullet$ and $s\bullet$ denote $cos(\bullet)$ and $sin(\bullet)$, respectively; $\boldsymbol{\eta} = [x, y, z]^T$; $\boldsymbol{v} = [u, v, w]^T$; $x$, $y$, $z$, $\theta$, and $\psi$ are the positions and orientations of the vehicle in the world coordinate; $u$, $v$, $w$, $q$, and $r$ are the translational velocities and angular velocities; $\tau_u$, $\tau_q$, and $\tau_r$ are the control inputs; $\tau_{eu}$, $\tau_{ev}$, $\tau_{ew}$, $\tau_{eq}$, $\tau_{eu}$, and $\tau_{er}$ are the bounded external disturbances, such as ocean currents and waves; $m_{ii}(i = 1, 2, 3, 5, 6)$ represent the terms for the combined mass and intertia parameters; $d_1 = (W - B)c\theta$; $d_2 = (z_g W - z_b B)s\theta$; $W$ is the gravity, $B$ is the buoyancy of the vehicle; $f_k(k)(k = u, v, w, q, r)$ represents the hydrodynamic damping and friction terms; and $\mathbf{R} \equiv \mathbf{R}_z(\psi)\mathbf{R}_y(\theta)$ is the rotation matrix of $\{B\}$ with respect to $\{W\}$. From Equation (2), the dynamics of the controllable states are rearranged as follows:

$$
\begin{aligned}
\overline{m}_u \dot{u} + h_u &= \tau_u, \\
\overline{m}_q \dot{q} + h_q &= \tau_q, \\
\overline{m}_r \dot{r} + h_r &= \tau_r,
\end{aligned}
\tag{3}
$$

where $\overline{m}_u$, $\overline{m}_q$, and $\overline{m}_r$ are the positive constants which represent the known ranges of inertia and $h_u$, $h_q$, and $h_r$ are nonlinear terms, defined as

$$
\begin{aligned}
h_u &\equiv (m_{11} - \overline{m}_u)\dot{u} - m_{22}vr + m_{33}wq + f_u(u)u + \tau_{eu}, \\
h_q &\equiv (m_{55} - \overline{m}_q)\dot{q} - (m_{33} - m_{11})uw + f_q(q)q + (d_2 + \tau_{eq}), \\
h_r &\equiv (m_{66} - \overline{m}_r)\dot{r} - (m_{11} - m_{22})uv + f_r(r)r + \tau_{er}.
\end{aligned}
\tag{4}
$$

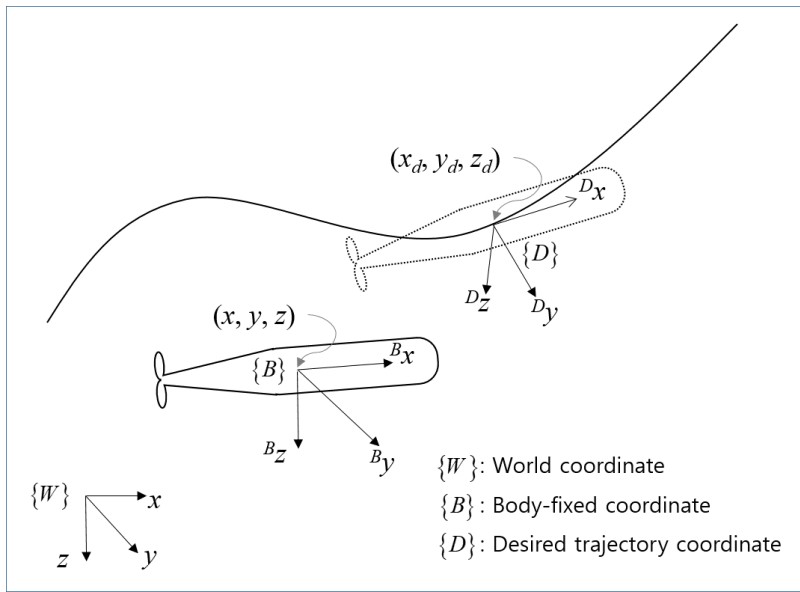

**Figure 3.** Definition of coordinates.

### 2.2. Desired Trajectory

Due to the underactuated nature, only three trajectory variables can be designed independently. Note that in Equation (2), there are only three independent control inputs to control five DOFs in the world coordinate. We can set the independent trajectory variables

as $x$, $y$, and $z$ in $\{W\}$. The trajectories can be represented by the following continuous time functions [8]:

$$\boldsymbol{\eta}_d = [x_d(t), y_d(t), z_d(t)]^T. \tag{5}$$

Note that, when taking into account Equation (1), the angular and velocity trajectories according to Equation (5) satisfy the following relationship [8]:

$$
\begin{aligned}
\theta_d &= -sin^{-1}\left(\dot{z}_d / \sqrt{\dot{x}_d^2 + \dot{y}_d^2 + \dot{z}_d^2}\right), \\
\psi_d &= atan2(\dot{y}_d, \dot{x}_d), \quad and, \\
u_d &= \sqrt{\dot{x}_d^2 + \dot{y}_d^2 + \dot{z}_d^2}, \\
q_d &= \dot{\theta}_d, \\
r_d &= \dot{\psi}_d c\theta_d.
\end{aligned}
\tag{6}
$$

The goal of this paper is to design a controller for an AUV system represented by Equations (1) and (3) to track the desired trajectories defined in Equations (5) and (6). The controller is specifically designed to reduce the tracking error along $\boldsymbol{\eta}_d$ by using control inputs.

### 2.3. Dynamics of Tracking Error

In this subsection, we arrange the tracking error dynamics of the vehicle by describing them in the desired trajectory coordinate $\{D\}$. This approach helps to minimize changes in the relationship between variables in different coordinates [38,39]. When there is no tracking error, $u$ affects $^Dx$ for every attitude of the vehicle, and the transformation can be easily achieved as follows [40]:

$$^D\boldsymbol{\eta} = \mathbf{R}_d^T\boldsymbol{\eta} - \mathbf{R}_d^T\boldsymbol{\eta}_d, \tag{7}$$

where $^D\boldsymbol{\eta} = \begin{bmatrix} ^Dx, {}^Dy, {}^Dz \end{bmatrix}^T$ denotes the translational position of the vehicle with respect to $\{D\}$ and $\mathbf{R}_d \equiv \mathbf{R}_z(\psi_d)\mathbf{R}_y(\theta_d)$. Using Equation (7), the desired trajectories in Equation (5) can be transformed into those in $\{D\}$ as follows:

$$^D\boldsymbol{\eta}_d = \mathbf{0}. \tag{8}$$

Subtracting Equation (7) from Equaiton (8) yields the following relationship:

$$^D\boldsymbol{\eta}_e = \mathbf{R}_d^T\boldsymbol{\eta}_e, \tag{9}$$

where $\bullet_e \equiv \bullet_d - \bullet$. Note that from Equation (9), the tracking problem in $\{D\}$ is identical to that in $\{W\}$. Convergence of the tracking error in $\{D\}$ guarantees convergence in $\{W\}$. This is because $\mathbf{R}_d$ is a rotation matrix and cannot be singular. By taking the derivatives of Equation (9) with Equations (1) and (5) and introducing positive constants $\bar{\alpha}_u$, $\bar{\alpha}_\psi$, and $\bar{\alpha}_\theta$, the error dynamics of the state variable in $\{D\}$ can be rearranged as follows [8]:

$$
\begin{bmatrix} ^D\dot{x}_e \\ ^D\dot{y}_e \\ ^D\dot{z}_e \end{bmatrix} = \begin{bmatrix} \bar{\alpha}_u u_e \\ \bar{\alpha}_\psi \psi_e \\ -\bar{\alpha}_\theta \theta_e \end{bmatrix} + \begin{bmatrix} \lambda_x \\ \lambda_y \\ \lambda_z \end{bmatrix}. \tag{10}
$$

Refer to [8] for a detailed derivation of Equation (10). The last term in the above equation represents the nonlinear terms defined as follows:

$$
\begin{bmatrix} \lambda_x \\ \lambda_y \\ \lambda_z \end{bmatrix} = -\begin{bmatrix} \bar{\alpha}_u u_e \\ \bar{\alpha}_\psi \psi_e \\ -\bar{\alpha}_\theta \theta_e \end{bmatrix} + \boldsymbol{v}_d - \mathbf{R}_d^T\mathbf{R}\boldsymbol{v} - \boldsymbol{\omega}_d^{\times D}\boldsymbol{\eta}_e, \tag{11}
$$

where $\boldsymbol{v}_d = [u_d, 0, 0]^T$ and $\boldsymbol{\omega}_d^\times$ is a skew-symmetric matrix of $\boldsymbol{\omega}_d \equiv [0, q_d, r_d]^T$ [8]. Aside from that, from Equations (1) and (6), the attitude error is obtained as follows:

$$
\begin{aligned}
\dot{\theta}_e &= q_e, \\
\dot{\psi}_e &= \bar{\alpha}_r r_e + \lambda_\psi,
\end{aligned}
\tag{12}
$$

where $\bar{\alpha}_r$ is a positive constant and

$$
\lambda_\psi = -\bar{\alpha}_r r_e + r_d / c\theta_d - r/c\theta.
\tag{13}
$$

As a result, the tracking error dynamics are expressed in Equations (10) and (12). The objective of this paper is to design a control input in Equation (3) to stabilize the error dynamics, particularly $^D x_e$, $^D y_e$, and $^D z_e$.

### 2.4. Controller Design Using the Coupled Error Dynamics and Time Delay Estimation

In this paper, our goal is to design the desired error dynamics for $u_e$, $r_e$, and $q_e$ that can asymptotically stabilize the tracking errors of $^D x_e$, $^D y_e$, and $^D z_e$, respectively. Note that from Equation (3), one can directly control the variables in $\{B\}$, namely $u$, $q$, and $r$, by designing appropriate control inputs $\tau_u$, $\tau_q$, and $\tau_r$, respectively. From Equations (10) and (12), $u$ adjusts $^D x_e$, $q$ affects $\theta_e$ and consequently $^D z_e$, and $r$ determines $\psi_e$ and therefore $^D y_e$. Thus, coupled error dynamics between the variables in $\{B\}$ and $\{D\}$ can be designed to stabilize the variables in $\{D\}$. The desired error dynamics of $u_e$, $r_e$, and $q_e$ are designed as follows:

$$
\begin{aligned}
\dot{u}_e + K_u u_e + K_x\,^D x_e &= 0, \\
\dot{q}_e + K_q q_e + K_\theta \theta_e - K_z\,^D z_e &= 0, \\
\dot{r}_e + K_r r_e + K_\psi \psi_e + K_y\,^D y_e &= 0,
\end{aligned}
\tag{14}
$$

where $K_\bullet > 0$ represents the control gains.

In the manner of the computed torque control, the controller for Equation (3) can be designed as follows:

$$
\begin{aligned}
\tau_u &= \widehat{h}_u + \overline{m}_u \mu_u, \\
\tau_v &= \widehat{h}_q + \overline{m}_q \mu_q, \\
\tau_r &= \widehat{h}_r + \overline{m}_r \mu_r,
\end{aligned}
\tag{15}
$$

where $\widehat{\bullet}$ denotes the estimate of $\bullet$ and $\mu_u$, $\mu_q$, and $\mu_r$ are the command inputs to insert the desired dynamics for $u$, $q$, and $r$, respectively. When $\widehat{\bullet} = \bullet$, the controlled dynamics is obtained from Equations (3) and (15) as follows:

$$
\begin{aligned}
\mu_u - \dot{u} &= 0, \\
\mu_q - \dot{q} &= 0, \\
\mu_r - \dot{r} &= 0.
\end{aligned}
\tag{16}
$$

To induce the desired error dynamics in Equations (14)–(16), the command inputs are designed as follows:

$$
\begin{aligned}
\mu_u &= \dot{u}_d + K_u u_e + K_{xp}\,^D x_e, \\
\mu_q &= \dot{q}_d + K_q q_e - K_\theta \theta_e - K_{zp}\,^D z_e, \\
\mu_r &= \dot{r}_d + K_r r_e + K_\psi \psi_e + K_{yp}\,^D y_e.
\end{aligned}
\tag{17}
$$

To implement the controller in Equation (15), it is necessary to obtain $\widehat{h}_u$, $\widehat{h}_q$, and $\widehat{h}_r$, the estimates of Equation (4). However, obtaining an exact dynamic model of Equation (4) is difficult and time-consuming. To address this, we employ the TDE [2,30,41,42] for robust and efficient estimation. The key idea behind TDE is that if the system dynamics are given as a continuous or piece-wise continuous function, then the variation in the dynamics during a very short time can be negligible. Thus, the value of the dynamics

at the current time can be estimated by using the value of the dynamics at a short time before. Based on this idea, the system dynamics can be estimated indirectly by utilizing previous information on the system input and output. From Equation (3), the dynamics can be estimated as follows:

$$
\begin{aligned}
\widehat{h}_{u(t)} &= h_{u(t-L)} = \tau_{u(t-L)} - \overline{m}_u \dot{u}_{(t-L)}, \\
\widehat{h}_{q(t)} &= h_{q(t-L)} = \tau_{q(t-L)} - \overline{m}_q \dot{q}_{(t-L)}, \\
\widehat{h}_{r(t)} &= h_{r(t-L)} = \tau_{r(t-L)} - \overline{m}_r \dot{r}_{(t-L)},
\end{aligned}
\tag{18}
$$

where $L$ denotes a short time delay which is commonly set as the sampling time of the control system. As a result, the final form of the proposed controller is Equation (15) with Equations (17) and (18).

It is noteworthy that the proposed controller in Equation (15) with Equations (17) and (18) only requires the first derivative of the state variables and the states themselves. The linear and angular velocities can be measured using IMU and DVL, while the vehicle's position can be obtained through a navigation algorithm such as a Kalman filter, utilizing sensors such as IMU, DVL, the depth sensor, and the digital compass [34]. The advantage of not requiring high-order differentiation of the states is that it helps stabilize the controller in experimental environments. This is because differentiating the states amplifies the noise effect present in the state measurements. In comparison, the BCTDE [2,7,8] necessitates third-order differentiation of the states. Therefore, one can expect that the proposed controller is relatively easier to stabilize in experimental environments. In addition, note that the TDE method does not require the entire vehicle dynamics model. One can design the proposed controller by only selecting the inertial gains, such as $\overline{m}_u$, $\overline{m}_q$, $\overline{m}_r$, $\overline{\alpha}_u$, $\overline{\alpha}_\psi$, $\overline{\alpha}_\theta$, and $\overline{\alpha}_r$, as well as the feedback gains, such as $K_x$, $K_u$, $K_y$, $K_\psi$, $K_r$, $K_z$, $K_\theta$, and $K_q$.

### 2.5. Error Dynamics of the Proposed Controller

The TDE provides an efficient way to estimate the nonlinear dynamics of the vehicle, but it cannot estimate the dynamics variation exactly during a sampling time $L$. Thus, an estimation error of the dynamics remains. Taking into account the estimation error of the TDE, one can rearrange the error dynamics in Equaiton (14) by utilizing Equations (3) and (15) with Equations (17) and (18) as follows:

$$
\begin{aligned}
\dot{u}_e + K_u u_e + K_x{}^D x_e &= \epsilon_u, \\
\dot{q}_e + K_q q_e + K_\theta \theta_e - K_z{}^D z_e &= \epsilon_q, \\
\dot{r}_e + K_r r_e + K_\psi \psi_e + K_y{}^D y_e &= \epsilon_r,
\end{aligned}
\tag{19}
$$

where $\epsilon_\bullet$ denotes the TDE errors, which are defined as follows:

$$
\begin{aligned}
\epsilon_u &\equiv \overline{m}_u^{-1} \left( h_{u(t)} - h_{u(t-L)} \right), \\
\epsilon_q &\equiv \overline{m}_q^{-1} \left( h_{q(t)} - h_{q(t-L)} \right), \\
\epsilon_r &\equiv \overline{m}_r^{-1} \left( h_{r(t)} - h_{r(t-L)} \right).
\end{aligned}
\tag{20}
$$

From Equations (10), (12), and (19), the error dynamics of the controlled system for the forward, lateral, and vertical directions, respectively, are as follows:

$$\begin{bmatrix} {}^D\dot{x}_e \\ \dot{u}_e \end{bmatrix} = \underbrace{\begin{bmatrix} 0 & \bar{\alpha}_u \\ -K_x & -K_u \end{bmatrix}}_{\mathbf{A}_x} \begin{bmatrix} {}^D x_e \\ u_e \end{bmatrix} + \underbrace{\begin{bmatrix} \lambda_x \\ \epsilon_u \end{bmatrix}}_{\mathbf{B}_x}, \tag{21a}$$

$$\begin{bmatrix} {}^D\dot{y}_e \\ \dot{\psi}_e \\ \dot{r}_e \end{bmatrix} = \underbrace{\begin{bmatrix} 0 & \bar{\alpha}_\psi & 0 \\ 0 & 0 & \bar{\alpha}_r \\ -K_y & -K_\psi & -K_r \end{bmatrix}}_{\mathbf{A}_y} \begin{bmatrix} {}^D y_e \\ \psi_e \\ r_e \end{bmatrix} + \underbrace{\begin{bmatrix} \lambda_y \\ \lambda_\psi \\ \epsilon_r \end{bmatrix}}_{\mathbf{B}_y}, \tag{21b}$$

$$\begin{bmatrix} {}^D\dot{z}_e \\ \dot{\theta}_e \\ \dot{q}_e \end{bmatrix} = \underbrace{\begin{bmatrix} 0 & -\bar{\alpha}_\theta & 0 \\ 0 & 0 & 1 \\ K_z & -K_\theta & -K_q \end{bmatrix}}_{\mathbf{A}_z} \begin{bmatrix} {}^D z_e \\ \theta_e \\ q_e \end{bmatrix} + \underbrace{\begin{bmatrix} \lambda_z \\ 0 \\ \epsilon_q \end{bmatrix}}_{\mathbf{B}_z}. \tag{21c}$$

By taking the Laplace transform of Equations (21a)–(21c), one can examine the stability of the error dynamics and the influence of the forcing functions $\mathbf{B}_x$, $\mathbf{B}_y$, and $\mathbf{B}_z$. The Laplace-transformed error dynamics can be obtained as follows:

$$\begin{bmatrix} {}^D x_e \\ u_e \end{bmatrix} = (s\mathbf{I} - \mathbf{A}_x)^{-1}\mathbf{B}_x = \frac{1}{|s\mathbf{I} - \mathbf{A}_x|} \begin{bmatrix} s + K_u & \bar{\alpha}_u \\ -K_x & s \end{bmatrix} \begin{bmatrix} \lambda_x \\ \epsilon_u \end{bmatrix}, \tag{22a}$$

$$\begin{bmatrix} {}^D y_e \\ \psi_e \\ r_e \end{bmatrix} = \frac{1}{|s\mathbf{I} - \mathbf{A}_y|} \begin{bmatrix} s^2 + K_r s + \bar{\alpha}_r K_\psi & \bar{\alpha}_\psi s + \bar{\alpha}_\psi K_r & \bar{\alpha}_\psi \bar{\alpha}_r \\ -\bar{\alpha}_r K_y & s^2 + K_r s & \bar{\alpha}_r s \\ K_y s & -K_\psi s - \bar{\alpha}_\psi K_y & s^2 \end{bmatrix} \begin{bmatrix} \lambda_y \\ \lambda_\psi \\ \epsilon_r \end{bmatrix}, \tag{22b}$$

$$\begin{bmatrix} {}^D z_e \\ \theta_e \\ q_e \end{bmatrix} = \frac{1}{|s\mathbf{I} - \mathbf{A}_z|} \begin{bmatrix} s^2 + K_q s + K_\theta & -\bar{\alpha}_\theta \\ K_z & s \\ K_z s & s^2 \end{bmatrix} \begin{bmatrix} \lambda_z \\ \epsilon_q \end{bmatrix}, \tag{22c}$$

where $s$ denotes the Laplace operator and

$$|s\mathbf{I} - \mathbf{A}_x| = s^2 + K_u s + \bar{\alpha}_u K_x, \tag{23a}$$

$$|s\mathbf{I} - \mathbf{A}_y| = s^3 + K_r s^2 + \bar{\alpha}_r K_\psi s + \bar{\alpha}_\psi \bar{\alpha}_r K_y, \tag{23b}$$

$$|s\mathbf{I} - \mathbf{A}_z| = s^3 + K_q s^2 + K_\theta s + \bar{\alpha}_\theta K_y. \tag{23c}$$

From Equations (22a)–(22c), the influence of each term of the forcing functions $\mathbf{B}_x$, $\mathbf{B}_y$, and $\mathbf{B}_z$ can be estimated. The tracking errors in a steady state can be analyzed by using the final value theorem of the Laplace transform: $\lim_{t\to\infty} x(t) = \lim_{s\to 0} s x(s)$. For example, from Equations (22a) and (23a), ${}^D x_e(s)/\lambda_x(s) = (s + K_u)/(s^2 + K_u s + \bar{\alpha}_u K_x)$. Assume that $\lambda_x(s)$ is given as a step function: $\lambda_x(s) = l/s$ with a constant $l$. Then, ${}^D x_e(t)|_{t\to\infty} = K_u l/(\bar{\alpha}_u K_x)$, and one can estimate that in the forward direction error, ${}^D x_e(t)$, there will be a steady state error dependent on the amount of disturbed dynamics $l$ and the control gains $\bar{\alpha}_u$, $K_u$, and $K_x$. The controller cannot perfectly compensate for the influence of the forcing function, but it can attenuate the influence by selecting appropriate control gains.

The characteristic equations in Equations (23a)–(23c) are useful for selecting appropriate control gains. In order to ensure stable error dynamics, the characteristic equations must satisfy the Hurwitz condition, and the gains $K_\bullet$ must be chosen accordingly. Additionally, the inertial gains $\bar{m}_u$, $\bar{m}_q$, $\bar{m}_r$, $\bar{\alpha}_u$, $\bar{\alpha}_\psi$, $\bar{\alpha}_\theta$, and $\bar{\alpha}_r$ can be obtained through tuning. Previous research works utilizing the TDE have suggested selecting inertial gains within a known range of the vehicle's inertial terms. If the vehicle model is unknown, however, then the inertial gains can be obtained through tuning [2,30].

## 3. Practical Issues for the Experiments

When setting up the controller for the experiment, practical issues such as the noise effects of the measurements and actuator limitations have to be considered. These issues will be discussed in the following subsections.

### 3.1. Handling the Noise Effect in the TDE

The TDE provides an effective and efficient method for estimating the nonlinear dynamics of vehicles. However, the use of state derivatives in the TDE amplifies the noise effect in the measurement of the state variables. In the case of underwater vehicles, the vehicle's position is usually estimated by Kalman filtering of the sensor data, such as acceleration from the IMU and velocity from the DVL, which have considerable measurement noise. The amplification of noise can undermine the stability conditions. One way to handle this is to use the low-pass filtering effects of the inertial gains of the TDE [30]. Decreasing the inertial gains shows a similar effect to low-pass filtering. However, we found experimentally that adjusting the inertial gains was not enough. Therefore, we devised a method to attenuate the noise effect of the TDE. The idea is quite simple: cut down the direct TDE value and supplement the remaining part with the averaged value of the TDE. Figure 4 shows the noise attenuation method in the TDE. Note that the use of an average filter in Figure 4 can attenuate the noise effect because the filter also averages the noise. However, the filter may slightly degrade the performance of the TDE because the filtered value of the TDE cannot estimate exactly any quick changes in vehicle dynamics. In the case of underwater vehicles, dynamic changes occur due to the vehicle dynamics and disturbance changes such as sea currents, which depend on the mission (or desired trajectory) and the environment. In the case of AUVs, dynamic changes may not be fast because they are commonly used for surveys of large areas, and disturbances such as sea currents change slowly according to tide variation.

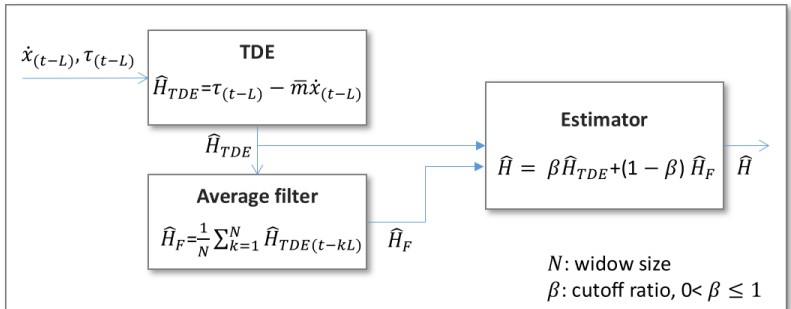

**Figure 4.** Filtering the TDE to reduce the noise effect. The algorithm reduces the direct TDE value and supplements the remaining part with the averaged value of the TDE. This mitigates the noise present in the TDE thanks to the averaging effect.

### 3.2. Handling Nonlinearity and the Limits of the Actuators

When designing a controller, it is important to consider the actuator characteristics, such as nonlinearity and the actuation limits. For instance, the thruster of a vehicle exhibits nonlinear dynamics between the propulsion force and rotation speed of the thruster. Moreover, the thruster has limits on rotational velocity and acceleration because the thruster is a mechanical system. In this subsection, we address compensation methods for the actuator characteristics, focusing on the thrusters in particular and briefly touching on the rudder fins and stern fins. The thruster dynamics are as follows [23,43]:

$$\tau_u = T_{|n|n}|n|n + T_{|n|u}|n|u, \tag{24}$$

where $n$ denotes the rotation velocity of the thruster and $T_{|n|n}$ and $T_{|n|u}$ represent the actuator coefficients corresponding to the rotational speed of the actuator and fluid speed around the actuator, respectively. By ignoring $T_{|n|u}|n|u$ and $T_{|n|n}$ from Equation (24), we adopted a simple thruster model, which is as follows:

$$\tau'_u = |n|n. \tag{25}$$

This is because the effect of $T_{|n|u}|n|u$, the term included in the RHS of Equation (24), can be compensated for by the feedback loop of the controller, and the scale coefficient $T_{|n|n}$ in Equation (24) can be adjusted by tuning the controller gain. Note that when substituting Equation (24) into the first equation of Equation (3), $T_{|n|u}|n|u$ can be treated as part of the nonlinear term $h_u$ (i.e., $h'_u = h_u - T_{|n|u}|n|u$). The coefficient $T_{|n|n}$ simply scales the value of $\overline{m}_u$ (i.e., $\overline{m}'_u = \overline{m}_u / T_{|n|n}$).

Regarding the actuation limit of the thrusting force, we considered the limits on the rotational velocity and acceleration of the thrusting propeller as follows:

$$\underline{n} = \begin{cases} n, & when \ |n| < n_{max}, \ |\dot{n}| < \dot{n}_{max}, \ n \geq 0, \\ f(n), & when \ any \ of \ above \ conditions \ is \ not \ satisfied, \end{cases} \tag{26}$$

where $\underline{n}$ represents $n$ with the actuation limit and $f(n)$ is a limiting function that considers the limit of $n$ and $\dot{n}$ as well as the sign of $n$. Note that in Equation (26), the sign condition $n \geq 0$ is included because only the forward thrusting force is available. In the case of the controllers using the TDE, handling the actuation limit to prevent the wind-up phenomenon is as straightforward as incorporating a limit block to ensure that the calculated control input for the TDE matches the actual value of the actuation applied to the vehicle [44]. Figure 5 shows the limiter block for handling the actuation limit of the thrusting force, while Figure 6 shows the structure of the limiter block, which is explained by Equation (26). In the case of the rudder fins and stern fins, one can handle the actuation limit in similar ways to the case of the thrusting force. The only differences are that (1) the actuation forces are linear with the fin motions [23] and (2) both positive and negative forces are available. By simply removing the condition $n \geq 0$ in Equation (26) and eliminating the blocks of sqrt() and square() in Figures 5 and 6, the same limiting algorithm can be applied to the rudder fins and the stern fins.

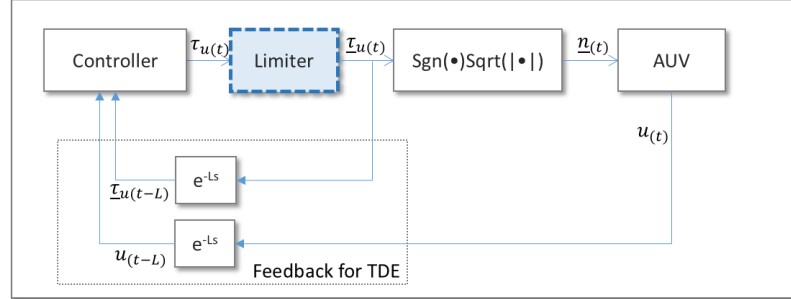

**Figure 5.** TDE feedback block for $h_u$ of the controller. $Sgn(\bullet)$ denotes a signum function, and $Sqrt(\bullet) \equiv \sqrt{\bullet}$. The 'Limiter' block is included to prevent the wind-up phenomenon due to the actuation limit [44]. The '$Sgn(\bullet)Sqrt(|\bullet|)$' block is for compensating for the actuation dynamics in Equation (25).

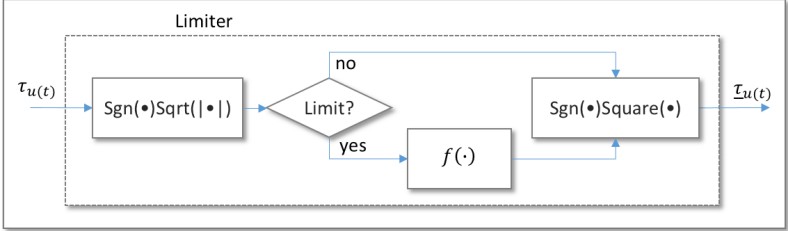

**Figure 6.** Limiter block of the thrusting force $\tau_u$: The thruster has actuation limits on the rotation speed and acceleration. Therefore, before checking the limitation, the '$Sgn(\bullet)Sqrt(|\bullet|)$' block is included to convert the control force $\tau_u$ into the rotation speed according to Equation (25).

## 4. Experimental Study

The tracking performance of the proposed controller was experimentally verified using the AUV platform depicted in Figure 2. The experiments were performed in the seawater at a port located in the South Sea of Korea. For the experiment, the control gains were set as follows. The inertial gains were set to $\overline{m}_u = 3000$, $\overline{m}_q = 0.7$, $\overline{m}_r = 1.0$, $\overline{\alpha}_u = 1.0$, $\overline{\alpha}_\psi = 1.5$, $\overline{\alpha}_\theta = 1.5$, and $\overline{\alpha}_r = 1.5$ by tuning, and the feedback gains $K_x = 1.0$, $K_u = 2.0$, $K_y = 0.216$, $K_\psi = 1.08$, $K_r = 1.8$, $K_z = 0.125$, $K_\theta = 0.75$, and $K_q = 1.5$ were selected for the desired error dynamics having poles at $p_{dx} = -1.0$ (double poles), $p_{dy} = -0.6$ (triple poles), and $p_{dz} = -0.5$ (triple poles). In this case, the characteristic equations in Equations (23a)–(23c) had poles at $p_{cx} = -1.0$ (double poles), $p_{cy} = -0.502, -0.649 \pm 0.740i$, and $p_{cz} = -0.897, -0.302 \pm 0.344i$, which were placed in the LHP. Regarding the noise-handling algorithm in Figure 4, the window size for the average filter was set at $N = 128$, and the $\beta$ values for $\widehat{h}_u$, $\widehat{h}_q$, and $\widehat{h}_r$ in Equation (18) were $\beta_u = 0.7$, $\beta_q = 0.5$, and $\beta_r = 0.9$, respectively.

### 4.1. Experimental Verification of the Noise-Handling Issue

In this subsection, the noise handling method described in Section 3.1 is experimentally verified. The experiments were conducted on the surface of the sea, with actuations in the XY plane. The thrusters and rudder fins were activated, while the stern fins were deactivated. The tracking performances were compared between the case where the noise-handling algorithm was not applied and the case where the algorithm was applied. A trajectory involving linear motion was used, as shown in Figure 7. The experimental results are presented in Figures 7–12. Figures 7–9 show the responses for the case without the noise-handling algorithm, while Figures 10–12 show the responses when the algorithm was applied. When comparing Figures 7 and 10, it may be difficult to recognize a significant difference in performance between the two cases. Aside from the initial errors in Figure 10, the error bounds appear to be similar in both cases. However, when comparing Figures 9 and 12, it is evident that the noise-handling algorithm was effective. In the case without the algorithm (Figure 9), the control input (rudder angle) switched frequently, resulting in erratic responses in $r$, $\psi$, and $^Dy$. This behavior was due to the TDE in Equation (18), which included the first-order derivative of the velocity state and amplified the noise present in the state measurement. In contrast, when the noise-handling algorithm was applied (Figure 12), the control input (rudder angle) reacted smoothly to tracking errors, leading to smooth convergence in the velocity and position responses. From Figures 8 and 11, it can be observed that the chatterings in the forward direction responses were alleviated when the noise-handling algorithm was adopted.

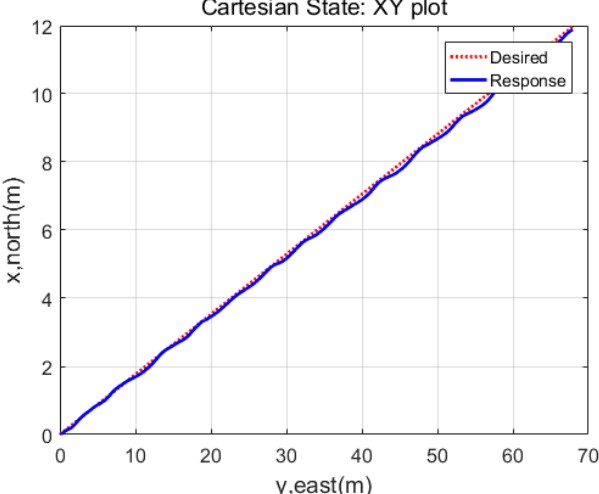

**Figure 7.** XY plot of the responses when noise-handling algorithm was not applied. The responses exhibited chattering behaviors due to the influence of measurement noise on the TDE.

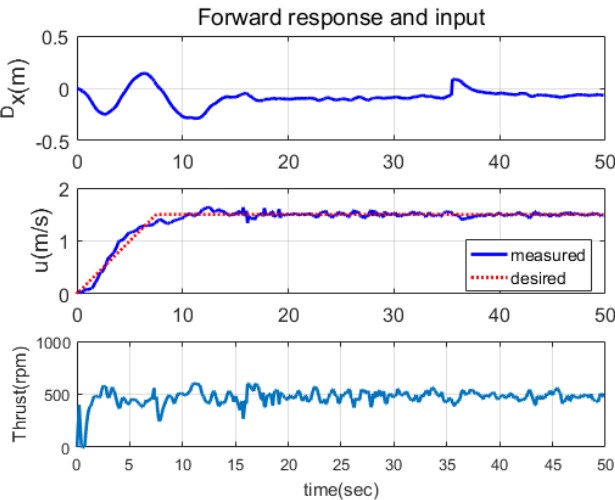

**Figure 8.** Forward direction responses when noise-handling algorithm was not applied. The responses were stable; however, there was some minor chattering in the control input.

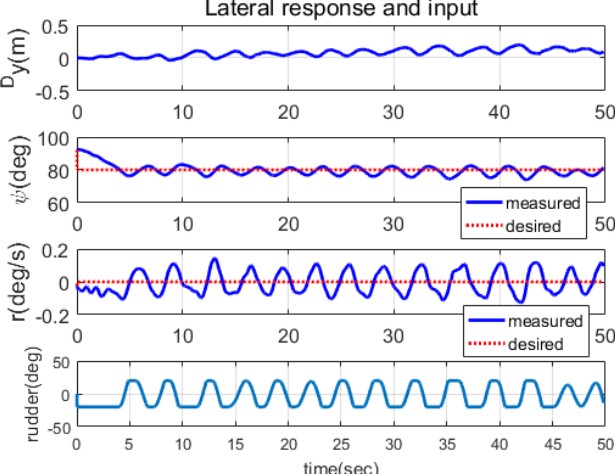

**Figure 9.** Lateral direction responses when noise-handling algorithm was not applied. The control input switched frequently, leading to chattering responses in $r$, $\psi$, and $^D y$.

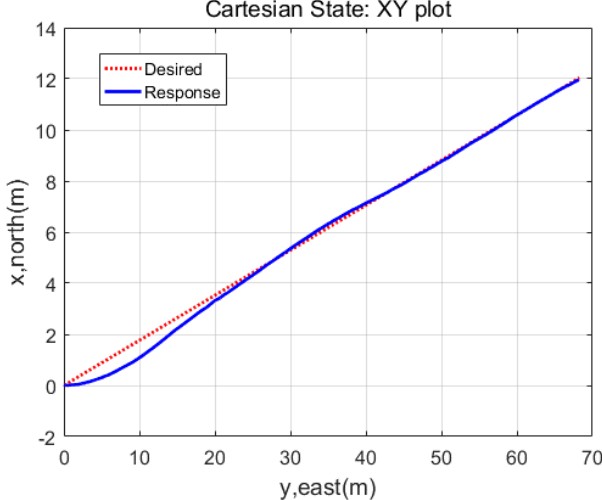

**Figure 10.** XY plot of the responses when noise-handling algorithm in Figure 4 was applied. The responses smoothly converged to the desired trajectories even in the presence of an initial angular error.

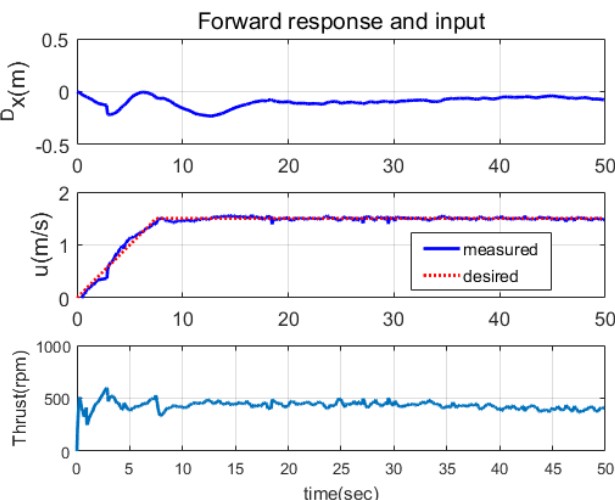

**Figure 11.** Forward direction responses when noise-handling algorithm in Figure 4 was applied. The responses exhibited smoother convergence compared with the responses shown in Figure 8.

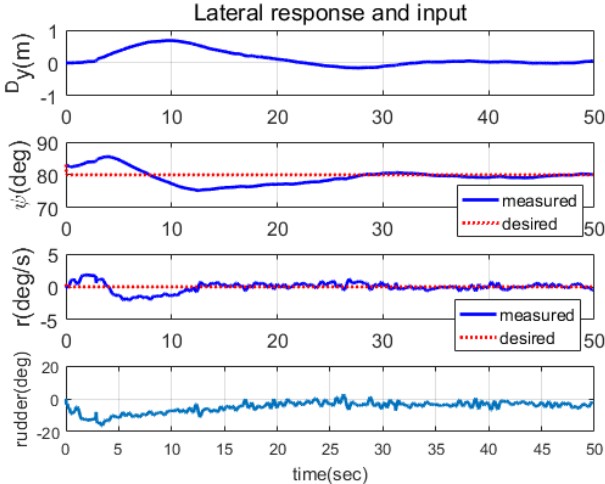

**Figure 12.** Lateral direction responses when noise-handling algorithm in Figure 4 was applied. The responses exhibited smoother convergence compared with the responses shown in Figure 9.

### 4.2. Experimental Verification of Tracking Performance in 3D Space Motion

The trajectory tracking performance of the proposed controller in 3D space motion was verified experimentally. As shown in Figure 13, the desired trajectory, drawing the shape of the number eight, was applied. The experimental results are presented in Figures 13–17. Figures 13 and 14 show the tracking responses in the world coordinate $\{W\}$, demonstrating that the controlled system had stable responses and followed the desired trajectory with bounded errors. The root mean square (RMS) errors in Table 1 indicate that the tracking performances were accurate enough to perform surveying missions requiring tracking errors of less than one meter. It can be observed from Table 1 that the RMS error in the vertical direction was slightly larger than those in the other directions due to buoyancy acting as a disturbance in the vertical direction. Figures 15–17 show the responses and control inputs for each direction. Note that in Figure 17, the response of $\theta$ exhibited a large tracking error induced by the controller to reduce the tracking error of $z$. In Figure 15, it can be observed that the tracking error of $^{D}x$ bounced around at times $t = 66$ s, 87 s, and 141 s, which was caused by irregular DVL signals resulting from stiff changes in the seabed.

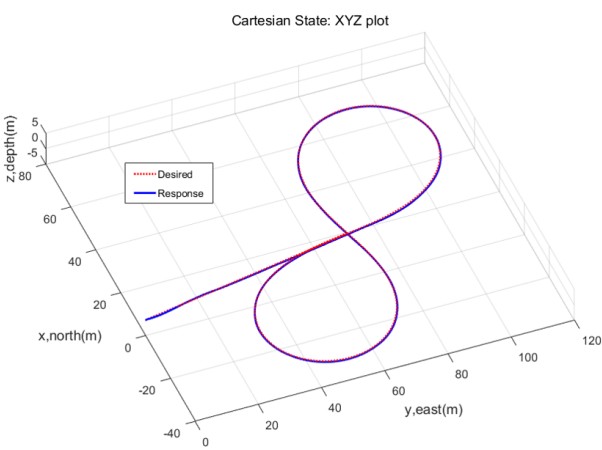

**Figure 13.** XYZ plot of the tracking responses.

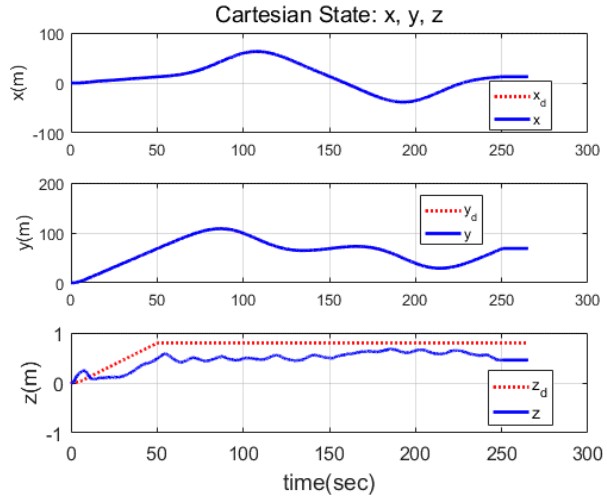

**Figure 14.** The time responses of the positions in a Cartesian space.

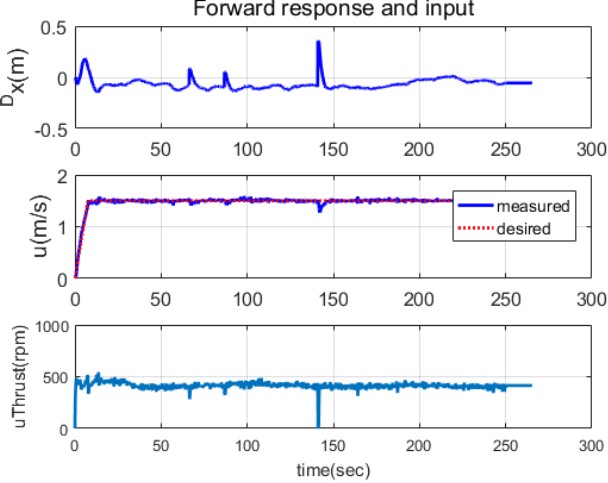

**Figure 15.** The responses and actuation in the forward direction. $^Dx$ converged to zero because, as explained in Equation (8), the desired trajectory expressed in the desired trajectory coordinate $\{D\}$ was zero. The responses of $^Dx$ and $u$ followed their desired trajectories smoothly, demonstrating the effectiveness of the noise filtering depicted in Figure 4. That aside, the tracking response of $^Dx$ bounced around at times $t = 66$ s, 87 s, and 141 s, which was caused by irregular DVL signals resulting from stiff changes in the seabed.

**Table 1.** The root mean square (RMS) errors in $\{D\}$.

| Direction | Forward | Lateral | Vertical | Total |
|---|---|---|---|---|
| RMS error (m) | 0.0838 | 0.1595 | 0.2663 | 0.3216 |

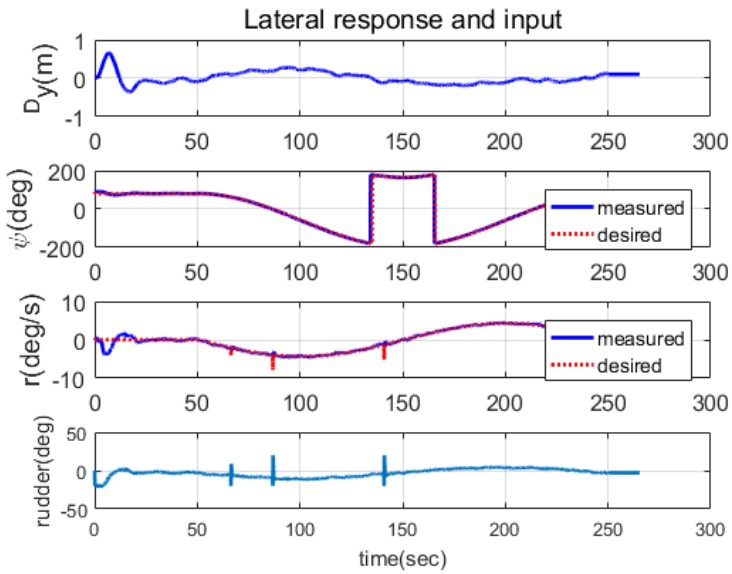

**Figure 16.** The responses and actuation in the lateral direction. The responses of $^{D}y$, $\psi$, and $r$ followed their desired trajectories smoothly, demonstrating the effectiveness of the noise filtering depicted in Figure 4. The responses did not exhibit any significant chattering, in contrast to the responses in Figure 9, which represents the case without noise filtering.

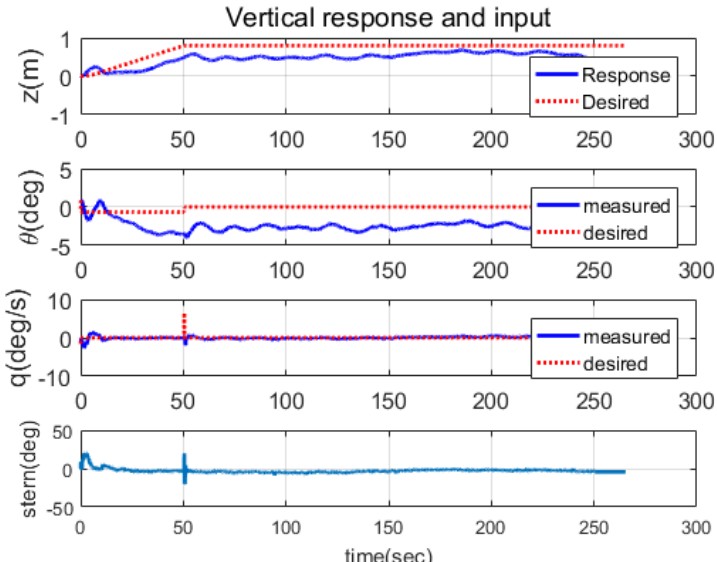

**Figure 17.** The responses and actuation in the vertical direction. The responses of $z$, $\theta$, and $q$ followed their desired trajectories smoothly. The response of $z$ showed a steady state error because the buoyancy acted as disturbances in the vertical direction. The error of $\theta$ induced a reduction in the error of $z$, corresponding to the desired error dynamics in (14). The responses having non-zero errors were matched with the error dynamics analysis in Equations (22a–22c), which explains that the proposed controller may not converge to zero if there are non-zero disturbances on the vehicle dynamics.

## 5. Conclusions

In this paper, we proposed a trajectory tracking controller for AUVs in 3D space motion, along with experimental verification on the sea. The main concept of the proposed controller is to design the desired error dynamics that combine the state variables in the body-fixed coordinate and the world coordinate (i.e., the desired trajectory coordinate) to address the underactuated nature of the vehicle. The TDE, an indirect estimation method utilizing control inputs and vehicle outputs, was employed to estimate the nonlinear dynamics and disturbances of the vehicle. Consequently, the proposed controller is relatively easy to implement as it does not require the entire dynamic model of the vehicle. In terms of experimental implementation, the controller is relatively easy to stabilize since it only requires the first derivatives of the states and the states themselves, thereby potentially mitigating noise amplification arising from differentiation. Practical issues related to implementation in experimental environments were also addressed. A noise-filtering algorithm for the TDE was developed, and compensation methods for the mechanical limitations and nonlinear dynamics of the actuators were devised. The performance of the proposed controller was validated through experiments in seawater. The experimental results demonstrate the effectiveness of the noise-filtering algorithm in stabilizing the control performance. Through trajectory tracking control experiments in 3D space motion, it was verified that the proposed controller achieves accurate tracking performance, rendering it suitable for survey missions requiring precise tracking performance with errors of less than one meter.

**Author Contributions:** Conceptualization, G.R.C. and J.-H.L.; methodology, G.R.C.; software, G.R.C. and H.K. (Hyungjoo Kang); validation, G.R.C., H.K. (Hyungjoo Kang), M.-G.K., M.-J.L., H.K. (Hosung Kim), H.L. and G.L.; formal analysis, G.R.C.; investigation, G.R.C.; resources, J.-H.L.; data curation, G.R.C.; writing—original draft preparation, G.R.C.; writing—review and editing, G.R.C.; visualization, G.R.C.; supervision, G.R.C.; project administration, G.R.C.; funding acquisition, G.R.C. All authors have read and agreed to the published version of the manuscript.

**Funding:** This research was supported by the Korea Institute of Marine Science & Technology Promotion (KIMST) and funded by the Ministry of Oceans and Fisheries of Korea (20220567, Development of standard manufacturing technology for marine leisure vessels and safety support robots for underwater leisure activities).

**Institutional Review Board Statement:** Not applicable.

**Informed Consent Statement:** Not applicable.

**Data Availability Statement:** Data are contained within the article.

**Conflicts of Interest:** The authors declare no conflict of interest.

## Abbreviations

The following abbreviations are used in this manuscript:

| | |
|---|---|
| AUV | Autonomous underwater vehicle |
| BC | Back-stepping control |
| BCTDE | Back-stepping control with time delay estimation |
| TDE | Time delay estimation |
| DOF | Degree of freedom |
| IMU | Inertial measurement unit |
| DVL | Doppler velocity log |
| GNSS | Global navigation satellite system |
| RMS | Root mean square |

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
