# Peer review of "An Experimental Study on Trajectory Tracking Control of Torpedo-like AUVs Using Coupled Error Dynamics"

_jmse, doi:10.3390/jmse11071334_

Round 1

Reviewer 1 Report

In this paper, the authors propose a trajectory-tracking controller for torpedo-like autonomous underwater vehicles (AUVs) with under-actuation characteristics. The proposed controller overcomes the under-actuation problem by designing the desired error dynamics as a coupled form using state variables in body-fixed and world coordinates. The paper has some contributions but I have the following comments.

Add some numerical results in the abstract.

The introduction is not stated properly.

Please do a proper literature review and add a section of related work to the paper. Afterward, add the contributions of the work in point at the end of the introduction section.

What is the motivation behind this work? What is the significance of this work?

There should be a section for the system model followed by problem formulation and proposed solution.

There is no synchronization in the paper.

Then there should be a section simulation and results. I mean you have to rename your section experimental study to this. Further, please add some more discussion for the results in this section for each plot to understand the reader.

Please add the proposed algorithm and discuss its complexity. 

There are grammatical mistakes and extra spaces throughout the paper. Please review and refine the whole manuscript in the revised version.

Author Response

Please find the attached file: answer to reviewer.

Reviewer 2 Report

The English should be improved. There are a number of incorrect phrases and words, and many sentences cannot be understood. At present, it is difficult to fully comprehend the motivation and arguments being made.

Author Response

(The authors gave the same response as above.)

Round 2

Reviewer 1 Report

I have no further comments except one.

Please write the contributions of your work at the end of the introduction section in points.

Quality is much better in the revised version of the paper.